# Kinetic Changes in B7 Costimulatory Molecules and IRF4 Expression in Human Dendritic Cells during LPS Exposure

**DOI:** 10.3390/biom12070955

**Published:** 2022-07-08

**Authors:** Henry Velazquez-Soto, Fernanda Real-San Miguel, Sonia Mayra Pérez-Tapia, María C. Jiménez-Martínez

**Affiliations:** 1Department of Immunology and Research Unit, Institute of Ophthalmology “Conde de Valenciana Foundation”, Mexico City 06800, Mexico; henry.velazquez@institutodeoftalmologia.org (H.V.-S.); fereal92@gmail.com (F.R.-S.M.); 2Unidad de Desarrollo e Investigación en Bioprocesos (UDIBI), Departamento de Inmunología, Escuela Nacional de Ciencias Biológicas, Instituto Politécnico Nacional, Mexico City 11340, Mexico; sperezt@udibi.com.mx; 3Department of Biochemistry, Faculty of Medicine, National Autonomous University of Mexico, Mexico City 04510, Mexico

**Keywords:** LPS, costimulatory molecules, dendritic cells, cytokines, PDL1, PDL2, CD86, ICOS-L, B7, soluble costimulatory molecules, IRF4

## Abstract

A key aspect of the inflammatory phenomenon is the involvement of costimulatory molecules expressed by antigen-presenting cells (APCs) and their ability to secrete cytokines to set instructions for an adaptive immune response and to generate tolerance or inflammation. In a novel integrative approach, we aimed to evaluate the kinetic expression of the membrane and soluble B7 costimulatory molecules CD86, ICOS-L, PDL1, PDL2, the transcription factor Interferon Regulatory Factor 4 (IRF4), and the cytokines produced by monocyte-derived dendritic cells (Mo-DCs) after challenging them with different concentrations of stimulation with *E. coli* lipopolysaccharide (LPS) for different lengths of time. Our results showed that the stimuli concentration and time of exposure to an antigen are key factors in modulating the dynamic expression pattern of membrane and soluble B7 molecules and cytokines.

## 1. Introduction

Dendritic cells (DCs) are considered to be the primary antigen-presenting cells (APCs) due to their ability to sense, capture, process, and present antigens to T cells. The initiation and polarization of adaptative immune responses are well orchestrated by a display of signals, including the expression of membrane-bound costimulatory molecules, soluble costimulatory molecules, and the secretion of cytokines by DCs [1,2].

Distinct families of costimulatory molecules have been described. The best known is the B7 family, belonging to the immunoglobulin superfamily (IgSF) [3]. This family of molecules comprises costimulatory molecules which promote activation in T cells, such as CD86 and ICOS-L, and coinhibitory molecules which regulate the tolerance and suppression of functions, such as PDL1 and PDL2 [4]. Interestingly, previous studies have proven that soluble forms of B7 molecules (sB7) can be detected in different tissues and supernatants of cell cultures [5,6,7,8]. This finding improves our understanding of the mechanism of action independently of cell contact interactions. However, it is still unclear whether a costimulatory or coinhibitory profile of sB7 molecules prevails under the steady or activation states of dendritic cells, and what signals are conveyed to the T cells.

Critical regulators at the transcriptional level for B7 molecules have been reported. Transcription factors for CD80, CD86, and ICOS-L include NF-kB and PU.1 [9,10,11,12]. Recently, a novel transcription factor, the Interferon Regulatory Factor 4 (IRF4), was reported to be related to the expression of PDL1 and PDL2 [13,14,15].

B7 costimulatory molecules have received attention due to their relevance in clinical conditions such as allergies, autoimmunity, cancer, and transplantation [16]. Nevertheless, little is known about the behavior of these molecules in terms of both counterparts (stimulation and inhibition) in physiological conditions, since most in vitro studies have evaluated only single molecules and single time points, or have focused on samples of patients with specific clinical conditions [17,18,19].

In addition to the expression of B7 molecules, autocrine-secreted cytokines play a relevant role in the expression of costimulatory molecules. Previous studies have suggested that IL-6 downregulates CD86- and HLA-associated molecules, impairing T-cell activation capacity in DCs [20]; meanwhile, TNF-α favors the expression of these molecules (in a viral context) [21]. On the other hand, IFN-γ has been shown to directly regulate PDL1 and PDL2 expression in melanoma cells [22]. However, no studies have evaluated the profiles of cytokine and costimulatory molecules integrally.

The present study aimed to explore the dynamics of B7 costimulatory molecules in an integrative way, encompassing the membrane-bound costimulatory molecules, the soluble B7 costimulatory molecules, and the cytokines found in the microenvironment. Our present model proposed Mo-DCs as archetype APCs due to their feasibility of obtaining a significant yield from peripheral blood mononuclear cells (PBMNCs), as reported in previous studies [23,24,25]. On the other hand, the use of LPS as a model antigen has been widely explored due to its capacity to activate innate immune cells and induce the expression of costimulatory molecules and cytokine secretion by DCs [26,27,28].

## 2. Materials and Methods

### 2.1. Healthy Donors

Healthy donors underwent clinical and immunological evaluations to eliminate the presence of any clinical conditions. Experimental subjects gave their written informed consent to participate in this research. This research project adhered to the ethical principles of the Declaration of Helsinki. The Scientific, Bioethics, and Biosafety Committees at the Institute of Ophthalmology, “Foundation Conde de Valenciana”, approved this project.

### 2.2. Mo-DC Generation and Culture In Vitro

Peripheral blood was obtained by venipuncture and collected in BD Vacutainer™ K_2_EDTA tubes. Peripheral blood mononuclear cells (PBMNCs) were obtained through differential centrifugation using Lymphoprep. PBMNCs were cultured in 6-well flat-bottomed culture plates for 2 h at 36 °C with a 5% CO_2_ atmosphere to allow for the attachment of adherent cells. After 2 h of culture, suspended cells were discarded. In every culture for each donor, before adding recombinant cytokines for differentiation, adherent cells were detached by adding 2%EDTA+Trypsin, and the purity of CD14+ cells was assessed through flow cytometry.

Adherent cells were plated at a density of 5 × 10^5^ cells per well. Human recombinant cytokines IL-4 and GMC-CSF were added at 50 ng/mL and 80 ng/mL, respectively. Cells were cultured for seven days. On the third day, cells received a change of half of the media with the same concentration of cytokines.

On day 7, the immunophenotype of the Mo-DCs was evaluated by flow cytometry, by measuring the percentage of CD11c+, HLA-DR+, and CD14+ cells (Appendix A).

### 2.3. In Vitro Stimulation of Mo-DCs

To explore the dynamics of expression for the membrane and soluble B7 costimulatory molecules, Mo-DCs from healthy donors were stimulated in vitro with 100 ng/mL, 1 µg/mL, or 10 µg/mL of LPS from *E. coli* strain OB111:B4 (Sigma-Aldrich, St. Louis, MO, USA) for 12, 24, or 48 h. Unstimulated Mo-DCs were used as a control.

Cells and supernatants were collected and centrifuged for 20 min at 1000× *g*. The supernatant was frozen at −80 °C to further determine the cytokines and the soluble costimulatory molecules. Immunostaining for flow cytometry was performed immediately after cell harvesting.

### 2.4. Evaluation of B7 Costimulatory Molecules and IRF4 Expression in Dendritic Cells by Flow Cytometry

After stimulation, cells were harvested by adding 2% EDTA+trypsin. Cells were washed twice with PBS-A. Membrane staining was performed using fluorescent-labeled monoclonal antibodies from BioLegend (San Diego, CA, USA), CD11c-FITC, HLA-DR-PeCy7, CD14-PercP, CD86-PE, ICOSL-APC, PDL1-BV421, and PDL2-APC, for 30 min at 4 °C.

After membrane staining, cells were fixed and permeabilized with D Cytofix/Cytoperm™ according to the manufacturer’s instructions. Then, cells were incubated with anti-IRF4-PE. Next, cells were washed twice with PBS-A, resuspended in 300 mL of PBS, and immediately acquired in a BD FACS Verse cytometer. Fluorescence Minus One (FMO) and the BD™ Anti-Mouse Ig, κ/Negative Control Compensation Particles Set (BD™ Comp Beads) were used as controls for fluorescence compensation.

### 2.5. Flow Cytometry Analysis Strategy

Flow cytometry analysis was performed using FlowJo™ Software Version 10.0, BD. The analysis consisted of gating single cells (FSC-H vs. FSC-A). Then, size and complexity events suggestive of Mo-DCs were gated (FSC-A vs. SSC-A), and HLA-DR+ cells were considered as Mo-DCs. Finally, the mean fluorescence intensity (MFI) from every costimulatory molecule in every condition was used to perform the statistical analysis (Appendix A).

### 2.6. Evaluation of Soluble B7 Costimulatory Molecules

The culture supernatant from stimulated Mo-DCs was assessed to determine soluble CD86, ICOS-L, PDL1, and PDL2. ELISA kits from Cloud-Clone Corp (Katy, TX, USA) were used according to the manufacturer’s instructions. The detection limits were as follows: CD86: 0.057 ng/mL, ICOS-L: 0.062 ng/mL, PDL1: 0.061 ng/mL and PDL2: 0.075 ng/mL.

### 2.7. Evaluation of Soluble Cytokines

The BD Cytometric Bead Array (CBA) Human Th1/Th2/Th17 Cytokine Kit (BD Biosciences, San Jose, CA, USA) was used according to the manufacturer’s instruction to determine soluble cytokine (IL-2, IL-6, IL-10, TNF, and IFN-γ) from the supernatants of stimulated Mo-DCs. Data were analyzed using FCAP Array™ Software version 3.0 (BD Biosciences, San Jose, CA, USA). The detection limits were as follows: IL-2: 2.6 ng/mL, IL-6: 2.4 ng/mL, IL-10:4.5 ng/mL, TNF-α: 3.8, and IFN-γ: 3.7 ng/mL.

### 2.8. Statistical Analysis

All experiments were performed independently at least three times in each condition. A statistical analysis was performed using GraphPad Prism software. The colorblind-proof design was used in our colored bar graphics. A normality test was performed on all data obtained from previous ANOVA and post hoc statistical analyses. A Kruskal–Wallis test for comparing three or more groups was used to determine statistical differences in membrane costimulatory molecule expression. A Tukey’s test was used to determine statistical differences in the soluble cytokine production experiments. A Dunnett’s test was used to compare statistical differences in soluble costimulatory molecule assays. Correlations were performed with the Pearson test. In all cases, a value of *p* < 0.05 was considered statistically significant.

## 3. Results

### 3.1. Membrane B7 Molecules Express Differentially according to the Time of Stimulation and the Concentration of LPS

To evaluate the effect of the concentration of LPS and the time of stimulation on the expression of membrane B7 costimulatory molecules, Mo-DCs were stimulated with 100 ng, 1 μg, or 10 μg of LPS for 12, 24, or 48 h.

The coinhibitory molecules PDL1 and PDL2 were found to increase their expression after 12 h of stimulation with all the concentrations evaluated. PDL1 cumulatively increased its expression over time. In contrast, the costimulatory molecules CD86 and ICOS-L increased their expression after 48 h of stimulation with any concentration of LPS (Figure 1). In another approach, we calculated the costimulatory molecules’ fold increase by dividing the averaged MFI of the different concentrations by the unstimulated condition (Figure 2). Our results show that CD86 increased three times only at 48 h of stimulation, ICOS-L increased once at any time evaluated, PDL1 showed a gradual fold increase over time, PDL2 increased its MFI once at 12 h and 0.5 times at 48 h after stimulation, and IRF4 showed no statistically significant changes.

### 3.2. IRF4 Expression Correlates with PDL2

The transcription factor IRF4 increases its expression at 12 h when Mo-DCs were stimulated with 100 ng of LPS. When stimulated with 1 or 10 µg, the increase was found to occur at 48 h. A Spearman’s test showed a correlation with PDL2 expression but not with PDL1 (Figure 3).

### 3.3. Mo-DCs Produce Cytokines in a Dose–Response Fashion

An association between cytokines and costimulatory molecules has been proposed. To elucidate whether cytokines exhibit similar behavior to membrane B7 costimulatory molecules, cytokines in the supernatant were evaluated under the same conditions of time and concentration. TNF-α, IL-6, and IL-10 were found to increase after short stimulation times (12 h). In contrast, IFN-γ and IL-2 increased preferentially at 48 h with any of the concentrations evaluated. Interestingly, higher concentrations of LPS led to higher concentrations of IFN-γ (Figure 4).

### 3.4. Cytokine Production Correlates with Costimulatory Molecule Expression

A correlation test was performed to evaluate the association between cytokines and costimulatory molecules, finding a positive and statistically significant correlation between IL-2 and CD86 and PDL1. IFN-γ showed a correlation with CD86 and PDL1 (Figure 5).

### 3.5. sPDL1 and sPDL2 Are the Predominant Soluble B7 Molecules in the Supernatant of Mo-DCs under a Steady State and LPS Stimulation

The supernatant from LPS-stimulated and unstimulated Mo-DCs was collected and assessed for the soluble forms of costimulatory molecules CD86, ICOS-L, PDL1, and PDL2. Our results showed sPDL1 and sPDL2 to have higher concentrations in all conditions evaluated in a constant 2:1 ratio compared with CD86. sICOS-L was not detected in any condition (Figure 6).

## 4. Discussion

Whether an inflammatory response is evoked depends on different extrinsic and intrinsic characteristics of the antigen and the host, which together define immunogenicity [29]. One key aspect of the inflammatory response is the initiation of the adaptive immune response, which depends on the intertwined interaction between APCs and T cells [30]. MHC/peptide–TCR interaction, costimulatory molecules’ ligation with receptors, and cytokine production are the main phenomena occurring in the immunological synapse [31]. As proposed by García and Ismail, in this interaction, not only the function but also the spatiotemporal interactions are critical for the signaling output [32]. Under these premises, questions have arisen regarding how the time of exposure and the concentration of the antigen can act as modulating factors that define the expression of costimulatory molecules, and, in turn, the activation of adaptative immunity. Langenkamp et al., using a similar approach, explored the effect of stimulation with different concentrations of LPS in dendritic cells over time, and found that the peak production of cytokines IL-6, TNF-α, IL-10, and IL-12 occurred at 12 h of stimulation. These authors showed that high antigen concentrations and a short period of stimulation favor Th1 polarization, while low antigen concentrations and long periods of stimulation lead to Th2 polarization [33].

Different studies have aimed to elucidate the role of costimulatory molecules; however, to the authors’ knowledge, not a single study has comprehensively evaluated costimulatory molecules and coinhibitory molecules in an established model. Some of the drawbacks of previous studies comprise the heterogeneity of the patients’ clinical conditions, the heterogeneity of the cell line evaluated, and a lack of standardized conditions (the length of time and antigen concentration used for stimulation). The present model provides novel insights for a better comprehension of the dynamic expression of B7 costimulatory molecules and their soluble forms. LPS was selected as the antigen for our model since multiple reports recommend it as an immunogenic biomolecule and report its costimulatory molecule induction activity [34,35,36].

According to our findings, we propose that B7 costimulatory molecules express differently in response to different antigen concentrations and stimulation times. Interestingly, we can observe the expression of the coinhibitory molecules PDL1 and PDL2 at an early stimulation time and under low antigen concentrations. Meanwhile, the costimulatory molecules CD86 and ICOS-L express at late times regardless of the antigen concentration. Similar to the previous results, increased production of TNF-α, IL-6, and IL-10 can be observed at early times of stimulation, which may influence the PDL1 and PDL2 expression. In contrast, cytokines related to T-cell activation, IFN-γ and IL-2, had a maximum secretion peak at the most prolonged time evaluated, similarly to the CD86 and ICOS-L expression. This differential kinetic expression could be explained by other reported biological mechanisms affecting protein expression, such as post-translational modifications and miRNA regulation, which should be addressed further to explain the B7 dynamic expression [37,38,39,40].

We also reaffirmed the proposed correlation between IRF4 and PDL2 [12,13,14], although further inhibition experiments must be performed to consistently prove this phenomenon. Finally, regarding how cytokines may influence the expression of costimulatory molecules, although using different conditions or cell populations, our findings are supported by previous experiments. Yokoseki demonstrated the effect of IFN-γ on the expression of CD86 in Langerhans cells. Meanwhile, Mimura and Qian reported on the IFN-γ–PDL1 axis in gastric cancer cell lines and glioma cells, respectively [41,42,43]. Concerning soluble forms of costimulatory molecules, we found PDL1 and PDL2 to be the predominant soluble B7 forms in Mo-DC cultures, regardless of stimuli conditions in the culture. It is important to note that one limitation of our work was the methodology used to obtain Mo-DCs. Nielsen et al. recently demonstrated that plastic adhesion, negative selection, and CD14^pos^ selection significantly impact the expression of some molecules, such as HLA, CD80, and CD163 [44]. According to these authors, plastic adhesion induces a higher expression of CD80 molecules, a member of the B7 family, and the cytokines TNF-α and IL-6 at 24 h in monocyte-derived macrophages. Nielsen’s report is particularly interesting, since the data shown in the present study could be regarded as having been influenced by the methodology used in our work (see Appendix A). Thus, our results should be interpreted with caution. In addition, future studies evaluating B7 molecules using isolation-specific techniques are necessary to discard the involvement of other cell populations that could affect the B7 kinetic findings reported here.

The data obtained in this study suggest that Mo-DCs may be predisposed to respond silently to, and prone to initiating, tolerogenic responses unless a threshold antigen concentration and stimulation time are reached, promoting B7 molecule activation and cytokine expression (see proposed integrative model, Figure 7).

Overall, these findings provide novel insights into dendritic cell behavior and allow for a better understanding of its role in promoting or halting adaptative immune responses.

Additionally, this useful information regarding costimulatory molecules, cytokines, and transcription factors establishes a framework for future exploration in different clinical situations, improving our understanding and leading to the development of therapeutic interventions.

## Figures and Tables

**Figure 1 biomolecules-12-00955-f001:**
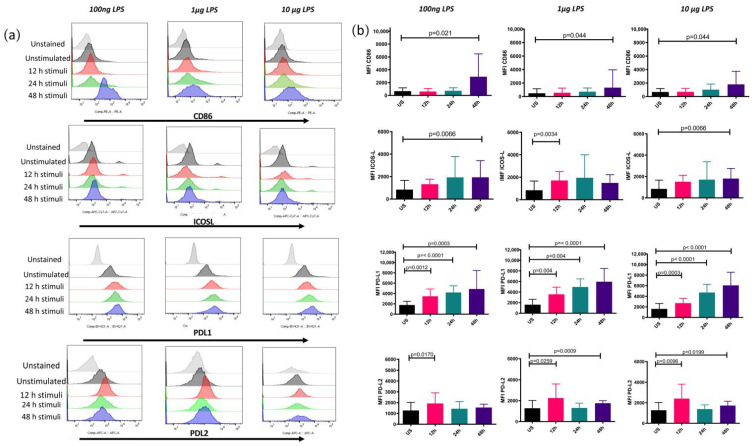
Costimulatory molecules show a differential kinetic expression. Mean fluorescence intensity for CD86, ICOS-L, PDL1, and PDL2 on Mo-DCs. Mo-DCs from healthy donors were stimulated for 12, 24, or 48 h with 100 ng, 1 μg, or 10 μg of LPS or left unstimulated (US). Cells were stained with fluorochrome-conjugated antibodies, as described in the materials and methods. (**a**) Representative histogram for each costimulatory molecule in the different experimental conditions evaluated. (**b**) The data shown represent the mean ± SD of MFI for 3 independent assays performed in triplicate.

**Figure 2 biomolecules-12-00955-f002:**
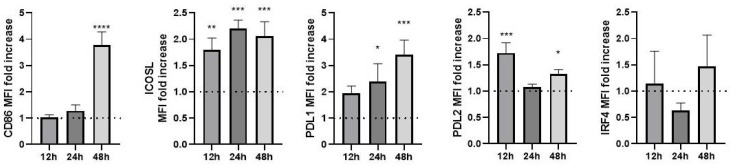
Fold increase in costimulatory molecules. The fold increase in costimulatory molecules and IRF4 MFI was calculated by dividing the averaged MFI obtained under the different concentrations of LPS stimuli by the MFI found in unstimulated Mo-DCs. The data shown represent the mean ± SD for 3 independent assays performed in triplicate. The dotted line represents basal expression. * *p* < 0.05, ** *p*<0.005, *** *p* < 0.0005, **** *p* < 0.0001.

**Figure 3 biomolecules-12-00955-f003:**
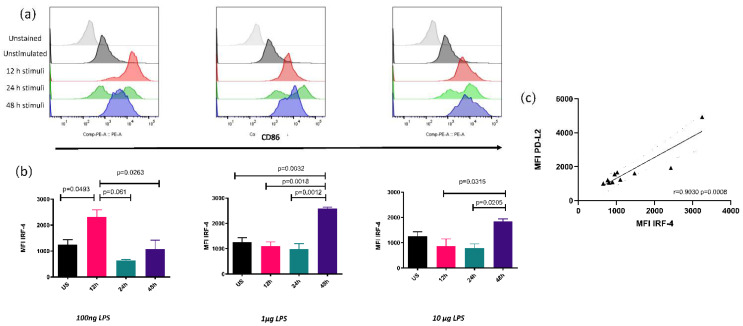
IRF4 shows a differential kinetic expression and correlates with PDL2 expression. (**a**) Representative histogram for IRF4 in different experimental conditions. (**b**) Mean of MFI ± SD for 3 independent assays performed in triplicate is depicted. (**c**) Pearson’s correlation between IRF4 and PDL2 expression. ▲ Represents individual correlation plots.

**Figure 4 biomolecules-12-00955-f004:**
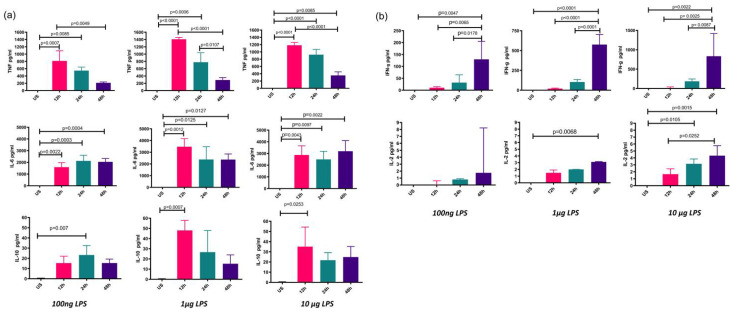
Soluble cytokine production kinetics. Mo-DCs from healthy donors were stimulated for 12, 24, or 48 h with 100 ng, 1 μg, or 10 μg of LPS or were left unstimulated (US). The supernatant was collected and assessed for soluble cytokines (**a**) TNF-α, IL-6, IL-10, (**b**) IFN-γ, and IL-2, as described in the material and methods. The mean ± SD of 3 independent assays performed in triplicate is depicted in the graphs.

**Figure 5 biomolecules-12-00955-f005:**
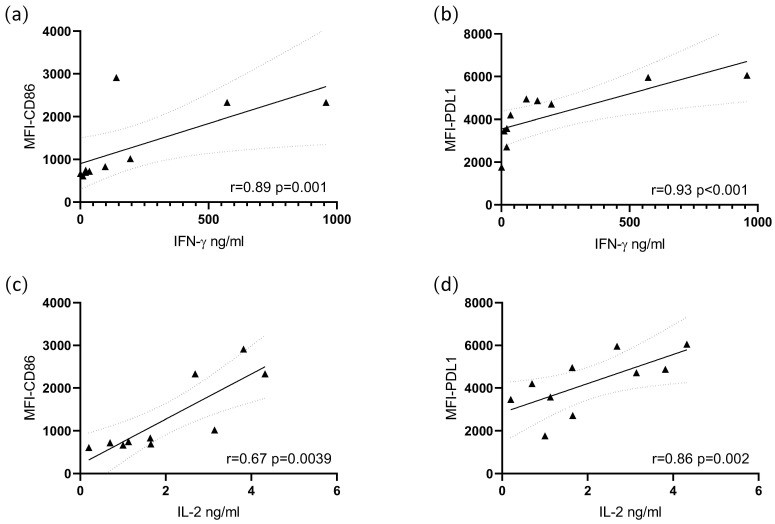
Correlation between cytokine production and costimulatory molecule expression. A Pearson correlation was performed by plotting the concentration of each cytokine vs. the MFI of each costimulatory molecule in the same experimental conditions. (**a**) IFNγ vs. CD86, (**b**) IFNγ vs. PDL1, (**c**) IL-2 vs. CD86, (**d**) IL-2 vs. PDL1. ▲ Represents individual correlation plots.

**Figure 6 biomolecules-12-00955-f006:**
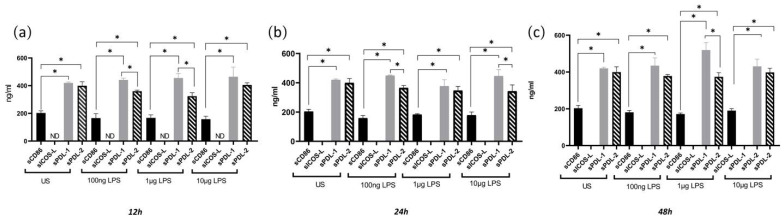
Concentration of soluble CD86, ICOS-L, PDL1, and PDL2 in the supernatant culture of Mo-DCs after (**a**) 12, (**b**) 24, or (**c**) 48 h of stimulation. Mo-DCs from healthy donors were stimulated for 12, 24, or 48 h with 100 ng, 1 μg, or 10 μg of LPS or were left unstimulated (US). ICOS-L was not detected in any of the conditions evaluated. The mean ± SD of soluble costimulatory molecule concentration for 3 independent assays performed in triplicate is depicted in the graphs. * *p* < 0.05.

**Figure 7 biomolecules-12-00955-f007:**
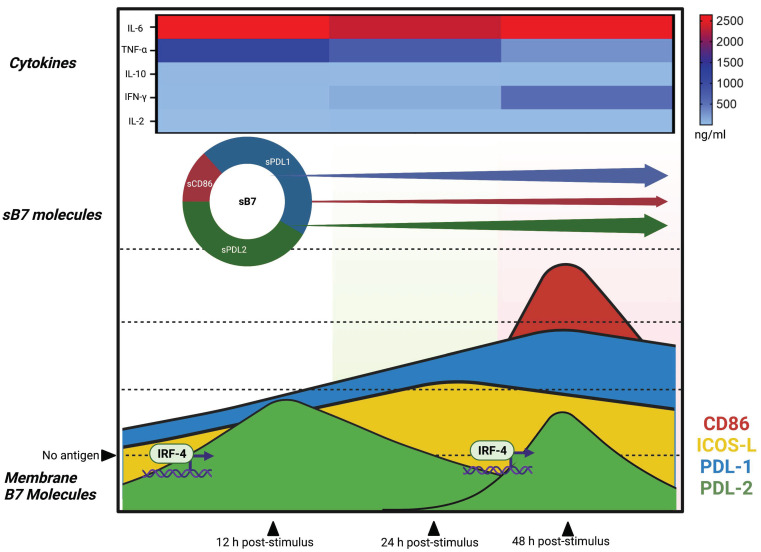
Integrative kinetic model explaining the behavior of B7 costimulatory molecules. Costimulatory molecules express in a differential fashion. Dendritic cells tend to increase PDL1 and ICOS-L expression during the early stages of antigen exposition. Additionally, transcription factor IRF4 expression promotes PDL2. After 24 h of stimulation, the expression of PDL1 is maintained. At 48 h, peak expression of molecules CD86 and PDL1 is observed. Regarding cytokines, TNF-α, IL-6, and IL-10 were found to increase at early stages; meanwhile, IFN-γ and IL-2 had a production peak at 48 h. Finally, soluble costimulatory molecules exhibited constant behavior regardless of the time of exposition or antigen concentration. sPDL1 and sPDL2 had twice the concentration of sCD86. Figure created with BioRender.com accessed on 28 June 2022.

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
