# Peer review of "Kinetic Changes in B7 Costimulatory Molecules and IRF4 Expression in Human Dendritic Cells during LPS Exposure"

_biomolecules, 2022, doi:10.3390/biom12070955_

Round 1

Reviewer 1 Report

Specific and major comments

In this study, authors explored the kinetic changes of B7 family proteins and IRF4 expression in human DCs in response to LPS stimuli. By using a monocyte-derived DC model, their findings showed that expression of B7 family protein CD86, ICOSL, PDL1 and PDL2 were dynamically changed in response to LPS stimuli in a dose-dependent and time-dependent manner. Moreover, this study also showed that IRF4 expression was differentially correlated with PDL2 expression in Mo-DCs in response to different LPS stimuli. Generally, this study has interest and merit, the experiments have been properly conducted, and the conclusions can be supported by the results. The major concerns and suggestions to improve the present manuscript are described underneath.

1. Why authors chose LPS at 0.1, 1, and 10 ug/mL but not at other doses for the investigation of B7 family protein expression and cytokine production by Mo-DCs? The reason or background may need further interpretation or introduction.

2. Authors focused on the time-dependent differences of the tested factors; however, the dose-responses of some of these factors such as CD86 at 48 h, IRF4 at 12 h and 48 h, and IL-10 at 12 h, are also of interest. Authors may further discuss the dose-dependent differences of these factors.

3. The introduction appears fragmented, and it is recommended to properly reorganized it.

4. Abbreviations should be well defined before their presentations. For example, “h” and hours in section 2. In addition, chemical formula should be correctly presented such as “K2EDTA” and “CO2” in section 2.2., and name of species such as “E. coli” should be italic.

5. As described in section 2.1., the number of human research approval by ethic committee should be provided.

6. The plot images in fig.1, fig.4 are not clear. Better images with high resolution should be provided.

Reviewer 2 Report

Paper by Velazquez-Soto et al. describe a classical approach to evaluate accessory molecules expression and secretion by plastic adherent cells from health subjects after LPS stimulation. This approach is well established and was very relevant in previous immunological researches (in particular in ‘90s years).

However there are some critical point that oriented researches towards other, well standardized,  methodologies.  Monocyte yield, subpopulation composition, Viability and Purity are strongly influenced by plastic adherence methodology and selection bias could affect the validity of results obtained (Read as an e.g. Nielsen el al. Immunology 2020 159:63-74. doi: 10.1111/imm.13125). Moreover, kinetic of stimulatory molecules expression and cytokines might be influenced by the time of recovery of Mo after plastic adherence and detaching stress.

On the other hand, I was unable to understand how many healthy PBMNC donors were involved. Actually there are reports indicating that monocyte response to antigens is variable from among different individuals. So to obtain reproducible results a congruous number of donors should be recruited, surely more than three and at least 10.

Regarding results reported, there are evidences in literature that, after transcription, PD-L1 and L2 are tightly regulated by miRNAs and RNA-binding proteins via the long 3'UTR. At translational level, proteins and their membrane presentation are tightly regulated by post-translational modification such as glycosylation and ubiquitination. Therefore, a better dissection of the regulation of PD-L1/PD-L2 kinetics and molecular interaction, possibly using molecular approach, would be necessary.

Minor:

Supplementary files result  unreadable
References should be update with most recent papers published on the topic that wold be considered in particular for discussion.

Reviewer 3 Report

The current study aims to document the kinetic changes in co-stimulatory molecules B7 and IRF-4 upon stimulation of CDs. The study is well designed and the results are clear and easily understood.

The only concern that authors may consider is focusing on the merit of their study and its significance, which can be added to the introduction.

Second, the authors should explain the roles of DCs and their role as professional APCs, defining the roles of co-stimulatory molecules and their significant roles. 

Minor, check E.coli should italic in all the manuscript.

Round 2

Reviewer 1 Report

Most of previous concerns have been solved and clarified. In addition, the manuscript is also properly improved.

Author Response

Minor changes were included in the new manuscript version according to other reviewers´ suggestions.

Reviewer 2 Report

Authors have improved quality of manuscript and added some information missed in the previous version.  However the methodological limitation of their experimental approach should better explained in discussion (page 8 line 326 and follow)

Reviewer 3 Report

The authors modified the manuscript and it is now suitable for publication.

Author Response

(The authors gave the same response as above.)
